# Deep Q-Network Based on a New Optimization Method with Applications to Path Planning

## Abstract

Path planning is an essential part for agents to navigate in complex environments efficiently. Recent advances in conventional methods and learning-based methods have improved adaptability in complex settings. However, balancing computational efficiency, optimality, and safety in different environments remains a critical open problem. In this paper, we propose a novel Q-Learning framework named OCPDQN based on the optimal control method with application to path planning problems. Furthermore, we improve OCPDQN by combining with Gauss-Newton and propose another new framework named GN-OCPDQN to avoid the extensive computation of the Hessian matrix. Compared to traditional deep Q-networks, which rely on the gradient descent method to update network parameters, the proposed methods present a faster convergence rate and higher robustness. The experimental results demonstrate that both OCPDQN and GN-OCPDQN frameworks show better learning performance than existing deep reinforcement learning methods in the path planning task.

## 1 Introduction

Path planning involves the computation of optimal and feasible paths from a start point to a goal under specific objectives and constraints (Karur et al., 2021). It plays a critical role in various applications such as robotics, autonomous transportation, and military operations, significantly reducing movement time and energy consumption through efficient route design.

Numerous classical approaches have been developed for path planning in both academic and industrial contexts. The A* algorithm (Li et al., 2022b) employs a heuristic search strategy to efficiently prune the search space while guaranteeing path optimality. Genetic algorithms (Alhijawi & Awajan, 2024) mimic mechanisms of natural selection and biological evolution to refine path solutions iteratively. Particle swarm optimization (Shami et al., 2022), inspired by collective swarm behavior, enables particles to collaboratively explore the solution space and share information to identify optimal paths.

Classical methods perform well in static environments but struggle in unknown, complex, and high-dimensional scenarios (Zhu et al., 2022). DRL (deep reinforcement learning), with its capabilities of autonomous learning, environmental adaptability, and long-term planning, overcomes the constraints of traditional algorithms and has emerged as a pivotal direction in the field of path planning (Zhang et al., 2022).

Deep reinforcement learning (DRL) integrates deep learning's perception with reinforcement learning's decision-making (Matsuo et al., 2022), enabling autonomous learning, environmental adaptation, and long-term planning for modern path planning (Qin et al., 2023). However, DRL still faces challenges during model optimization. Gradient descent, despite its widespread adoption, is notably susceptible to local optima and highly sensitive to the choice of learning rate. These limitations often necessitate a large number of iterations during network training, thereby impairing the overall convergence capability of the model (Zhang et al., 2024).

This paper introduces OCPDQN, a novel Q-learning framework for path planning that incorporates a super-linearly convergent optimization method. We innovatively integrate deep Q-networks (DQN) with techniques from optimal control problems (OCP) (Zhang et al., 2024), significantly enhancing the efficiency of policy learning and optimization. To eliminate the need for Hessian matrix com-

putations inherent in OCP, we further propose GN-OCPDQN, which leverages Gauss-Newton approximations, drastically accelerating computation—particularly in high-dimensional settings. Experimental evaluations show that OCPDQN achieves 47% faster convergence than DQN and 40% faster than PPO, while GN-OCPDQN delivers 41% and 33% improvements, respectively. These frameworks offer effective new solutions to complex path planning challenges.

We make the following contributions:

- Two novel frameworks, OCPDQN and GN-OCPDQN, are proposed to solve the path planning problem.

- We leverage the OCP method to significantly reduce the number of iterations during the training of the DQN network.

- The super-linear convergence rate of both frameworks is proven.

## 2 RELATED WORK

### 2.1 CONVENTIONAL METHODS FOR PATH PLANNING

Classical algorithms have demonstrated strong performance in deterministic global path planning. Genetic algorithms (Alhijawi & Awajan, 2024) simulate natural evolution through selection, crossover, and mutation operations. The Rapidly-exploring Random Tree (RRT) algorithm (Yu et al., 2024) grows tree-like structures from start points, with subsequent variants including Heuristic RRT (Hu et al., 2025) and Informed RRT* (Wu et al., 2024). Methods such as Elastic Bands (Amundsen et al., 2024), Artificial Potential Fields (APF) (Xie et al., 2022), and Vector Field Histogram (VFH) (Mohammed et al., 2024) are typically employed as local planners due to their rapid response to new information. Despite their historical success, traditional path planning algorithms become less effective as environmental complexity increases.

### 2.2 DRL IN PATH PLANNING

Compared with traditional path planning methods, DRL can adapt to complex environments. Li et al. (2022a) integrated DQN with Dueling DQN to address the end path planning problem of end-to-end intelligent driving vehicles. To guide the unmanned surface vehicle to the target area, Xiaofei et al. (2022) applied the Double-DQN algorithm to the global static path planning problem of amphibious unmanned vehicles. Gu et al. (2023) proposed the DM-DQN algorithm to improve the performance of DQN, and used it to train an agent in complex environments. There is extensive research on training autonomous vehicles using DQN, and almost all of them have achieved remarkable results.

Despite the remarkable performance of DQN and the improved DQN in addressing the path planning problems, the number of iterations required in the existing achievements is still large, which is insufficient to solve some practical path planning problems.

## 3 PROBLEM FORMULATION AND PRELIMINARIES

In this section, we formally describe the path planning problem in symbolic form first, and then present the definitions of the DQN network architecture. Finally, we elaborate on the OCP method.

### 3.1 PROBLEM DESCRIPTION OF PATH PLANNING

Path planning involves finding a feasible path from start $S$ to goal $G$ in environment $\mathcal{E}$ with obstacles $\mathcal{O}$, while optimizing criteria like minimal distance or time. Formally, let the environment be represented as $\mathcal{E}$, the start and target positions be $S$ and $G$ respectively, and the set of obstacles be $\mathcal{O}$. The planning algorithm seeks a path $P = \{p_0, p_1, ..., p_n\}$ such that $p_0 = S$, $p_n = G$, and $p_i \notin \mathcal{O}$ for all $i$. The solution must also satisfy any additional kinematic or dynamic constraints imposed by the agent.

## 3.2 THE ARCHITECTURE OF DQN

### 3.2.1 MDP MODEL

We formulate the path planning problem using a finite Markov Decision Process (MDP) over discrete time steps $t = 1, 2, \ldots, T$. The MDP is characterized by the tuple $(S, A, R, f, \gamma)$, representing the state space, action space, immediate reward, state transition model, and discount factor, respectively. The definitions of these elements for our algorithm are provided below.

- State Space: The state space $S$ consists of all possible states that the agent may occupy in the environment, which includes the position and obstacle distribution on a two-dimensional grid, and the actual coordinates of the agent. The size of the state space vector dimension is 2*11*11, that is, 242 dimensions.

- Action Space: In our algorithm, this contains eight parts: up, down, left, right, and the four diagonal directions.

- State Transition: The transition from state $s_t$ to the next state $s_{t+1}$ is defined as $s_{t+1} = \mathcal{F}(s_t, c_t, a_t)$. When the agent takes action $a_t$, its state will be updated from $s_t$ to $s_{t+1}$.

- Reward Function: Our reward function is designed to reflect potential events encountered by the agent during navigation. Specifically, the agent receives a reward of $+20$ for reaching the goal and a penalty of $-20$ for colliding with an obstacle. To promote the acquisition of short and smooth paths, we assign a penalty of $-1$ for movements along the four cardinal directions (forward, backward, left, right) and $-1.5$ for diagonal motions.

- Discount Factor: The discount factor is set to 0.99.

### 3.2.2 LOSS FUNCTION

In DQN, Q-function is defined as $Q(s, a; \theta)$, and the iterative form of the DQN update formula is given by $Q(s_t, a_t) = Q(s_t, a_t) + \alpha \left[ r_t + \gamma \max_{a'} Q(s_{t+1}, a') - Q(s_t, a_t) \right]$. When dealing with non-linear systems, the Q-function is typically modeled in a parameterized form. Therefore, it is necessary to rewrite the actor network in this form as follows,

$$\pi^t(x_k) = \pi^t(x_k; \theta_a^t) = \theta_a^t \, \psi(x_k), \tag{1}$$

where $\pi(x_k; \theta_a^t)$ represents the outputs of the actor networks, $x_k$ denotes the current state of the agent at the $k$-th iteration, $\theta_a^t$ denotes the parameters of the network weight matrix for the actor networks, and $\psi(\cdot)$ represents the vectors of activation functions.

The objective of DQN is to minimize the TD error between the current Q-value and the target Q-value $\mathcal{L}(\theta_a) = \mathbb{E}(s, a, r, s') \sim D \left[ (Q\theta_a(s, a) - y)^2 \right]$, where the target Q-value $y$ is defined as $y = r + \gamma \cdot \max_{a'} Q_{\theta_c}(s', a')$. In the traditional DQN algorithm, the Adam optimizer is used to perform gradient descent on $\theta$ $\theta_a \leftarrow \theta_a - \alpha \cdot \nabla_{\theta_a} \mathcal{L}(\theta_a)$. The optimal parameter $\theta_a^*$ is defined as the value of $\theta_a$ that minimizes the loss function $\theta_a^* = \arg \min_{\theta_a} \mathcal{L}(\theta_a)$.

### 3.2.3 NEURAL NETWORK

We train the agent using the OCPDQN and GN-OCPDQN frameworks. The network input consists of the obstacle distribution on a 2D grid and the agent's coordinates, while the output produces scores for eight possible actions.

The model's weight matrices are randomly initialized and optimized via gradient descent. At each step, the agent processes state $s_t$ through two convolutional and pooling layers, reshapes the hidden output to a 1D vector, and passes it through two fully connected layers to obtain action scores. The agent selects action $a_t$ with the highest score, updates its position, observes reward $r_t$, and transitions to new state $s_{t+1}$.

## 3.3 OCP METHOD

The OCP method represents a recent advancement in optimization techniques (Zhang et al., 2024). By reformulating the original optimization problem as an optimal control problem, the OCP method structures the iterative update process to minimize losses in future time steps.

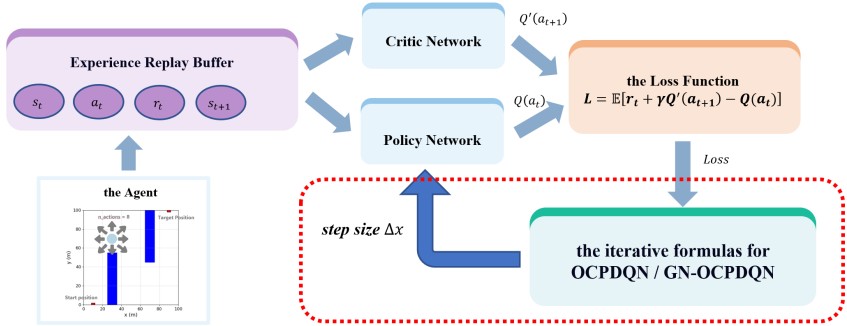

Figure 1: Efficient agent training framework with OCPDQN/GN-OCPDQN algorithm

Consider a twice differentiable function $f(x) : \mathbb{R}^n \mapsto \mathbb{R}^1$. The objective of this nonlinear optimization problem is to find the minimizer of $f(x)$. We rewrite the optimization problem as follows.

$$\min_u \sum_{i=0}^{N} \left[ f(x_i) + \frac{1}{2} u_i^T R u_i \right] + f(x_{N+1}), \tag{2}$$

$$\text{subject to } x_{i+1} = x_i + u_i, \tag{3}$$

where $x_i \in R^n$ and $u_i \in R^n$ are the state and control of system (1), respectively, integer $N > 0$ is the control terminal time, and positive definite matrix $R$ is the control weight matrix. The speed of convergence in the OCP method is influenced by the control matrix $R$. As the value of $R$ becomes larger, the convergence rate slows down.

Based on the reference (Zhang et al., 2024), we can get the iterative formula as follows:

$$x_{i+1} = x_i - z_i(x_i), \tag{4}$$

$$z_i(x_i) = (R + f''(x_i))^{-1} \left[ f'(x_i) + R z_{i-1}(x_i) \right], \tag{5}$$

$$z_0(x_i) = (R + f''(x_i))^{-1} f'(x_i), \tag{6}$$

where $z_i(x_i)$ is the step size for updating the parameter $x_i$ at each iteration. We can find the minima of $f(x)$ by this OCP method.

## 4 METHODOLOGY

To handle the issue of path planning problems, we propose a novel framework named OCPDQN. In addition, to avoid the expensive cost of computing the Hessian matrix, we further propose an algorithm named GN-OCPDQN.

### 4.1 OCPDQN METHOD

#### 4.1.1 UPDATING FORMULAS

Based on the OCPDQN method we propose, the iterative update formulas for the network parameters can be expressed as follows:

$$\theta_{a(i+1)} = \theta_{a(i)} - \overline{g}_i(\theta_{a(i)}), \tag{7}$$

$$\overline{g}_i(\theta_{a(i)}) = \left( R + \nabla^2 \mathcal{L}(\theta_{a(i)}) \right)^{-1} \times \left[ \nabla \mathcal{L}(\theta_{a(i)}) + R \overline{g}_{i-1}(\theta_{a(i)}) \right], \tag{8}$$

$$\overline{g}_0(\theta_{a(i)}) = \left( R + \nabla^2 \mathcal{L}(\theta_{a(i)}) \right)^{-1} \nabla \mathcal{L}(\theta_{a(i)}), \tag{9}$$

where $\theta_{a(i)}$ denotes the value of the network parameter $\theta_a$ at the $i$-th update iteration, and $\overline{g}_i(\theta_{a(i)})$ represents the step size for updating the parameter $\theta_{a(i)}$ in this iteration.

### 4.1.2 ALGORITHM PROCESS

In this part, we present the pseudo-code representation of the OCPDQN. At each step, the agent selects an action via $\epsilon$-greedy policy, executes it, and stores the experience $(s, a, s', r, done)$ in the replay buffer $\mathcal{D}$. If $|\mathcal{D}| < B$, the update is skipped; otherwise, a batch is sampled, loss $\mathcal{L}$ is computed, and parameter update $\delta$ is obtained using Eq. (7-9). The target network updates every $T$ episodes.

---

**Algorithm 1:** OCPDQN Algorithm

---

**Input:** Environment, batch size $B$, target update interval $T$
**Output:** Optimized policy network
1 Initialize environment, policy network $\pi$, target network $\pi_{\text{tar}}$, replay buffer $\mathcal{D}$;
2 **for** $ep \leftarrow 1$ *to* $N$ **do**
3    Reset environment, obtain initial state $s_0$;
4    $done \leftarrow$ False, $cum\_reward \leftarrow 0$, $step \leftarrow 0$;
5    **while** *not done and* $step < L$ **do**
6       Select action $a$ using $\epsilon$-greedy policy from $\pi(s)$;
7       Execute $a$, observe $s', r, done$;
8       Store transition $(s, a, s', r, done)$ in $\mathcal{D}$;
9       **if** $|\mathcal{D}| \geq B$ **then**
10          Sample batch $\mathcal{B}$ of size $B$ from $\mathcal{D}$;
11          Compute loss $\mathcal{L}$;
12          Update $\pi$ using gradient from $\mathcal{L}$ (Eq. 7-9);
13       $s \leftarrow s'$, $cum\_reward \leftarrow cum\_reward + r$, $step \leftarrow step + 1$;
14    **if** $ep \bmod T = 0$ **then**
15       Update $\pi_{\text{tar}} \leftarrow \pi$;

---

### 4.1.3 CONVERGENCE ANALYSIS

The weights of the actor network $\theta_a$ of OCPDQN can converge to the optimal weight $\theta_{a*}$ at a super-linear rate. Based on reference (Zhang et al., 2024) and Lemma 1 in the Appendix, the following relationship can be obtained.

$$
\begin{aligned}
\theta_{a(i+1)} - \theta_{a*} &= \theta_{a(i)} - \theta_{a*} - \overline{g}_i(\theta_{a(i)}) \\
&= \theta_{a(i)} - \theta_{a*} - [\overline{g}_i(\theta_{a*}) + \overline{g}'_i(\theta_{a*})(\theta_{a(i)} - \theta_{a*}) + o(|\theta_{a(i)} - \theta_{a*}|)] \\
&\approx \theta_{a(i)} - \theta_{a*} - [\overline{g}'_i(\theta_{a*})(\theta_{a(i)} - \theta_{a*})] \\
&= (I - \overline{g}'_i(\theta_{a*}))(\theta_{a(i)} - \theta_{a*}) \\
&= \left(R + \nabla^2\mathcal{L}(\theta_{a*})^{-1}R\right)^{i+1}\left(\theta_{a(i)} - \theta_{a*}\right)
\end{aligned}
\tag{10}
$$

where $o(|\theta_{a,c(i)} - \theta_{a*}|)$ is the higher order approximation term. Thus we have

$$
\theta_{a(i+1)} - \theta_{a*} \leqslant \left(R + \nabla^2\mathcal{L}(\theta_{a*})^{-1}R\right)^{i+1}\left(\theta_{a(i)} - \theta_{a*}\right).
\tag{11}
$$

When $\nabla^2\mathcal{L}(\theta_{a*}) > 0$, we have $\left\|\left(R + \nabla^2\mathcal{L}(\theta_{a*})\right)^{-1}R\right\| < 1$. When the number of iterations $i$ is sufficiently large, $\left(R + \nabla^2\mathcal{L}(\theta_{a*})^{-1}R\right)^{i+1} \to 0$, and thus we have

$$
\theta_{a(i+1)} - \theta_{a*} = \left(R + \nabla^2\mathcal{L}(\theta_{a*})^{-1}R\right)^{i+1}\left(\theta_{a(i)} - \theta_{a*}\right) = 0.
\tag{12}
$$

Hence, it can be seen that the actor network of OCPDQN ultimately converges to the optimal solution at a super-linear rate. The detailed proof is in the Appendix section.

## 4.2 GN-OCPDQN METHOD

### 4.2.1 IMPROVED UPDATING FORMULAS

In OCPDQN, from Eq. (8), we can know that as the number of parameters increases, the dimension of the Hessian matrix grows rapidly, and the computational resources of OCPDQN needed also

increase significantly. Therefore, to accelerate the computation speed of the algorithm, we propose an improved algorithm, GN-OCPDQN in this part, which combines the Gauss-Newton method (Hao et al., 2024).

The Gauss-Newton method avoids the direct computation of the Hessian matrix by approximating it with the product of the Jacobian matrix (Hao et al., 2024). Specifically, for a nonlinear least squares problem with the loss function

$$L(\theta) = \frac{1}{2}\|r(\theta)\|^2 = \frac{1}{2}\sum_{i=1}^{n} r_i(\theta)^2, \tag{13}$$

where $r(\theta)$ is the residual vector, the gradient and Hessian of $L(\theta)$ can be written as

$$\nabla_\theta L(\theta) = J_r(\theta)^\top r(\theta), \nabla_\theta^2 L(\theta) = J_r(\theta)^\top J_r(\theta) + \sum_{i=1}^{n} r_i(\theta)\nabla_\theta^2 r_i(\theta), \tag{14}$$

where $J_r(\theta)$ is the Jacobian matrix of $r(\theta)$ with respect to $\theta$. The Gauss-Newton method neglects the second term involving the second derivatives of the residuals, and thus approximates the Hessian as

$$\nabla_\theta^2 L(\theta) \approx J_r(\theta)^\top J_r(\theta). \tag{15}$$

This approximation eliminates the need to compute the full Hessian matrix and only requires the computation of the Jacobian, which is typically much more efficient. In the following part, we replace the Hessian matrix in the OCP iterative formula with an approximation using the Jacobian matrix based on Eq. (13-15). During the training process of the DQN network, the loss function can be rewritten in the form of a nonlinear least squares problem, where the residual is defined as

$$r(\theta_a) = Q_{\theta_a}(s,a) - y. \tag{16}$$

Based on Eq. (13-15), the first and second derivatives of the loss function can be rewritten as follows:

$$\nabla_{\theta_a}\mathcal{L}(\theta_a) = \frac{\partial r(\theta_a)}{\partial \theta_a}^\top r(\theta_a), \nabla_{\theta_a}^2\mathcal{L}(\theta_a) = \frac{\partial r(\theta_a)}{\partial \theta_a}^\top \frac{\partial r(\theta_a)}{\partial \theta_a}, \tag{17}$$

where $\frac{\partial r(\theta_a)}{\partial \theta_a}$ denotes the Jacobian matrix of the residuals $r(\theta_a)$ with respect to $\theta_a$. Let $h(\theta_a) = \frac{\partial r(\theta_a)}{\partial \theta_a}$. Then, the iterative optimization formulas of the OCP method can be simplified as follows:

$$\theta_{a(i+1)} = \theta_{a(i)} - \hat{g}_i(\theta_{a(i)}), \tag{18}$$

$$\hat{g}_i(\theta_{a(i)}) = \left(R + h(\theta_{a(i)})^\top h(\theta_{a(i)})\right)^{-1} \times \left[h(\theta_{a(i)})^\top r(\theta_{a(i)}) + R\hat{g}_{i-1}(\theta_{a(i)})\right], \tag{19}$$

$$\hat{g}_0(\theta_{a(i)}) = \left(R + h(\theta_{a(i)})^\top h(\theta_{a(i)})\right)^{-1} h(\theta_{a(i)})^\top r(\theta_{a(i)}), \tag{20}$$

where $\hat{g}_i(\theta_{a(i)})$ represents the step size for updating the parameter $\theta_{a(i)}$ in this iteration.

### 4.2.2 ALGORITHM PROCESS

In this part, we present the pseudo-code of GN-OCPDQN. The training procedure of GN-OCPDQN utilizes Eq. (18-20) to update the network, thereby avoiding the explicit computation of the full Hessian matrix.

### 4.2.3 CONVERGENCE ANALYSIS

For parameters $\theta_{a(i)}$ based on Eq. (10-12), there holds

$$\theta_{a(i+1)} - \theta_{a*} \leqslant \left(R + J_r(\theta_{a*})^\top J_r(\theta_{a*})^{-1}R\right)^{i+1}\left(\theta_{a(i)} - \theta_{a*}\right). \tag{21}$$

Similarly, we have $J_r(\theta_{a*})^\top J_r(\theta_{a*}) > 0$, so $\left\|\left(R + J_r(\theta_{a*})^\top J_r(\theta_{a*})\right)^{-1}R\right\| < 1$. When the number of iterations $i$ is sufficiently large, $\left(R + \left(J_r(\theta_{a*})^\top J_r(\theta_{a*})\right)^{-1}R\right)^{i+1} \to 0$, we have

$$\theta_{a(i+1)} - \theta_{a*} = \left(R + (J_r(\theta_{a*})^\top J_r(\theta_{a*}))^{-1}R\right)^{i+1} \times \left(\theta_{a(i)} - \theta_{a*}\right) = 0. \tag{22}$$

Thus the weights of the actor network $\theta_a$ of GN-OCPDQN can converge to the optimal weight $\theta_{a*}$ at a super-linear rate. A more detailed proof is provided in the Appendix.

---

**Algorithm 2:** GN-OCPDQN Algorithm

---

**Input:** Environment, batch size $B$, target update interval $T$
**Output:** Optimized policy network

1 Initialize environment, policy network $\pi$, target network $\pi_{\text{tar}}$, replay buffer $\mathcal{D}$;
2 **for** $ep \leftarrow 1$ *to* $N$ **do**
3      Reset environment, obtain initial state $s_0$;
4      $done \leftarrow$ False, $cum\_reward \leftarrow 0$, $step \leftarrow 0$;
5      **while** *not done and* $step < L$ **do**
6          Select action $a$ using $\epsilon$-greedy policy from $\pi(s)$;
7          Execute $a$, observe $s', r, done$;
8          Store transition $(s, a, s', r, done)$ in $\mathcal{D}$;
9          **if** $|\mathcal{D}| \geq B$ **then**
10             Sample batch $\mathcal{B}$ of size $B$ from $\mathcal{D}$;
11             Compute loss $\mathcal{L}$;
12             Update $\pi$ using gradient from $\mathcal{L}$ (Eq.18-20);
13          $s \leftarrow s'$, $cum\_reward \leftarrow cum\_reward + r$, $step \leftarrow step + 1$;
14      **if** $ep \bmod T = 0$ **then**
15          Update $\pi_{\text{tar}} \leftarrow \pi$;

---

## 5 EXPERIMENTS

In this section, we apply the proposed frameworks to a path planning problem to demonstrate the training efficiency and convergence performance using PPO (Xiao et al., 2023) as the baseline. The experiments were carried out on a workstation with an Intel Xeon Gold 6348 CPU, 100GB RAM, and an NVIDIA A800 GPU, implemented using PyTorch. Besides, we compare the step time with the original DQN.

### 5.1 IMPLEMENTATION DETAILS

This paper designs a specialized 2D path planning map, spanning $100 \times 100$ meters with obstacles represented by several blue rectangular regions. The agent navigates from start $(10, 0)$ to goal $(90, 100)$ while avoiding obstacles, forming an S-curve optimal path.

During training, the path was discretized into continuous points through thousands of iterations. The agent begins each episode at the start position and concludes upon reaching the goal. An $\epsilon$-greedy strategy ($\epsilon$ decaying from 1 to 0.1) encourages exploration with eight possible actions (cardinal and diagonal directions).

To verify the effectiveness of OCPDQN and GN-OCPDQN, we select a representative method from the two types of classic reinforcement learning algorithms as the comparative methods. One is the original DQN proposed in (Wu & Suh, 2024), and the other is the PPO algorithm presented in (Wen et al., 2021), which belongs to the category of policy gradient. They achieve great performance in path planning for autonomous driving (Wu & Suh, 2024), (Wen et al., 2021), (Li et al., 2022a), (Xiao et al., 2023). Thus, we use the two models as the comparative methods.

### 5.2 LEARNING PERFORMANCE

This paper compares the learning performance of OCPDQN and GN-OCPDQN against traditional DQN and PPO algorithms. Step-reward curves demonstrate the training efficiency, while the number of steps required for path completion provides additional evaluation in the path planning scenario. This provides a comprehensive evaluation of the algorithms' learning performance.

#### 5.2.1 LEARNING PERFORMANCE OF OCPDQN

As shown in Figure 2 (a), the OCPDQN quickly improves its cumulative reward and converges to near-optimal performance in fewer episodes, while maintaining small fluctuations throughout

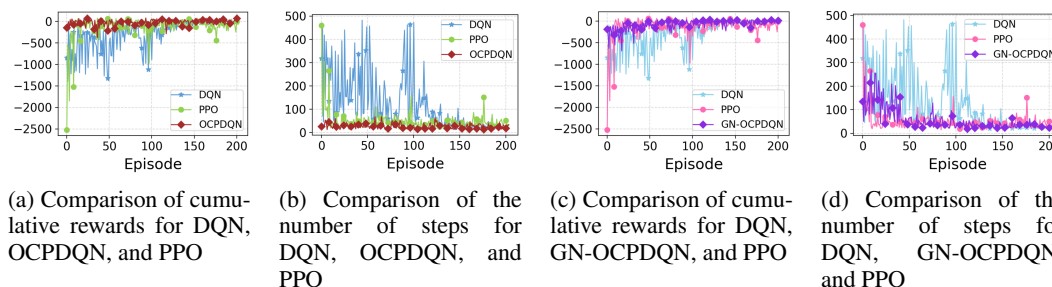

(a) Comparison of cumulative rewards for DQN, OCPDQN, and PPO

(b) Comparison of the number of steps for DQN, OCPDQN, and PPO

(c) Comparison of cumulative rewards for DQN, GN-OCPDQN, and PPO

(d) Comparison of the number of steps for DQN, GN-OCPDQN, and PPO

Figure 2: Performance comparison of different algorithms

training. By contrast, DQN exhibits much larger fluctuations and slower convergence. Although the PPO algorithm enables the agent to achieve relatively high rewards in the early stages of learning, the curve of the PPO algorithm exhibits great fluctuations in the subsequent training process.

Figure 2 (a) shows that while PPO occasionally achieves higher rewards (e.g., at episodes 50 and 75) due to exploratory randomness, these gains are unstable—evidenced by significant fluctuations even around episode 200. In contrast, OCPDQN converges more rapidly and stably to the optimal solution despite occasional early inferiority.

As seen in Figure 2 (b), OCPDQN quickly reduces path length to a stable 30 steps, while DQN converges gradually only after 100 episodes. Although PPO learns rapidly initially, its performance remains volatile over time.

Joint analysis of (a) and (b) confirms that OCPDQN enables faster convergence to higher rewards and more stable optimal paths, demonstrating significantly improved learning efficiency.

### 5.2.2 LEARNING PERFORMANCE OF GN-OCPDQN

As shown in Figure 2 (c), GN-OCPDQN achieves higher rewards faster than DQN and PPO, converging after 75 episodes. While DQN attains only low rewards by episode 120, and PPO learns shorter paths earlier (13 episodes), its unstable performance with fluctuating curves is outperformed by GN-OCPDQN's stability.

Although GN-OCPDQN's Gauss-Newton approximation initially lags behind PPO before episode 60, it converges stably by episode 90, whereas PPO continues fluctuating. Around the 100th episode, the training of the agent trained using GN-OCPDQN can be terminated. However, to highlight the advantages of GN-OCPDQN, we also let this agent train for 200 episodes, further demonstrating that the GN-OCPDQN framework we propose not only enables the agent to achieve higher rewards but also results in more stable learning performance.

### 5.2.3 COMPARISON OF OPTIMAL PATHS

Figure 3 illustrates the experimental outcomes of the agent's path-planning performance employing the DQN, OCPDQN, and GN-OCPDQN algorithms, respectively. All three methods successfully generate feasible paths that avoid obstacles from the start point to the goal. The DQN agent converges to a 21-step path after exhibiting considerable training instability, whereas the OCPDQN method achieves a shorter path of 18 steps with markedly improved convergence speed. By integrating the observations from Figure 3 with those from Figure 2 (c)(d), it is evident that GN-OCPDQN attains superior reward values within fewer training iterations and identifies a collision-free path consisting of 26 steps. Although this path is marginally longer than that obtained by DQN, the significantly accelerated convergence rate of GN-OCPDQN renders this outcome highly acceptable.

### 5.3 REAL-TIME PERFORMANCE

Real-time performance is a crucial metric in path planning, which refers to the time required to obtain an output decision from input data. The horizontal axis *steps* in Figures 4 represents each train-

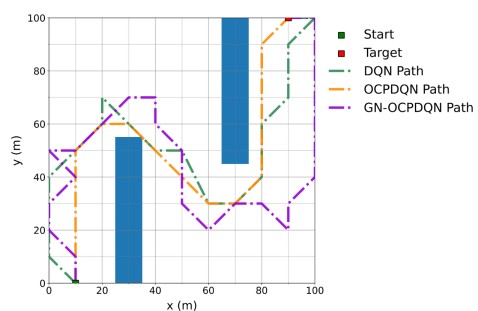

Figure 3: Comparison of the path of different algorithms

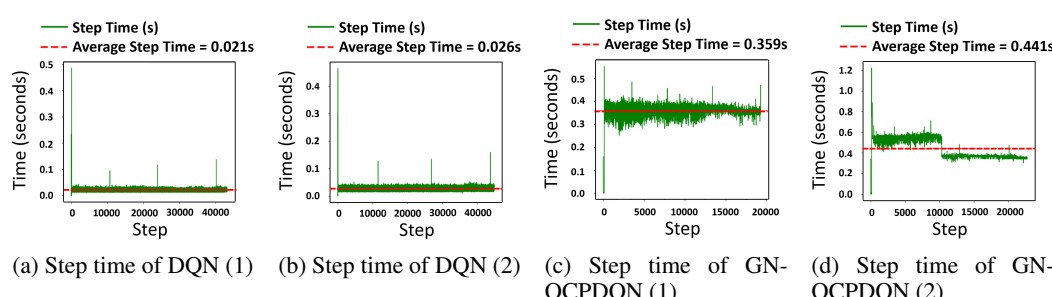

(a) Step time of DQN (1)  (b) Step time of DQN (2)  (c) Step time of GN-OCPDQN (1)  (d) Step time of GN-OCPDQN (2)

Figure 4: Step time comparison between DQN and GN-OCPDQN algorithms

ing step where the agent receives a movement command and updates the network parameters. As shown in Figures 4, GN-OCPDQN requires approximately 0.35 seconds per step—slightly longer than gradient descent methods. However, GN-OCPDQN converges in significantly fewer steps than DQN with gradient descent, as evidenced in Figures 2. Additionally, the resulting path is relatively short. Thus, the marginally longer computation time per step is acceptable given the substantial reduction in total steps and competitive path quality.

## 6 CONCLUSION AND DISCUSSION

This paper introduces two novel frameworks, OCPDQN and GN-OCPDQN, each with distinct characteristics. OCPDQN employs iterative formulas that leverage the Hessian matrix of network parameters to compute the updating step size, resulting in accelerated convergence. In contrast, GN-OCPDQN adopts the Gauss–Newton method to approximate the gradient, circumventing Hessian computations altogether. While this approximation entails a minor loss of accuracy, it markedly enhances the efficiency of step size calculation.

Based on their distinct characteristics, the two algorithmic frameworks excel in different application scenarios. When the network has a relatively small number of parameters (e.g., on the order of 1000), OCPDQN achieves faster convergence to the optimum than GN-OCPDQN. In contrast, for large-scale networks, GN-OCPDQN offers significantly higher computational efficiency, enabling quicker convergence and demonstrating superior scalability in high-dimensional parameter spaces.

## 7 REPRODUCIBILITY STATEMENT

To facilitate reproducibility and further research, the source code and implementation details for this work have been made publicly available at the following anonymized repository: `https://anonymous.4open.science/r/2026_anonymous-FF85`.

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

# A  Appendix

In this part, we will prove that the OCPDQN and GN-OCPDQN can converge to the optimal solution at a super-linear rate.

## A.1  Convergence Analysis of OCPDQN

The training objective of a neural network is to minimize the following function:

$$\min_{\theta} L(\theta) = \mathbb{E}_{(s,a,r,s')\sim D}\left[(Q_{\theta_a}(s,a) - y)^2\right]. \tag{23}$$

In OCPDQN, we propose an updated relation for (23) based on Eq. (10) in the main text

$$\min_{u} \sum_{i=0}^{N}\left[L(\theta_{a(i)}) + \frac{1}{2}u_i^T R u_i\right] + L(\theta_{a(N+1)})), \tag{24}$$

$$\text{subject to } \theta_{a(i+1)} = \theta_{a(i)} + u_i. \tag{25}$$

Thus, we can get the updating formulas of OCPDQN written as Eq. (7-9) in the main text.

**Lemma 1.** *Since $\theta_{a*}$ is the minimum value of the function $L(\theta)$, we have $\nabla L(\theta_{a*}) = 0$, $\nabla_{\theta}^2 L(\theta_{a*}) > 0$. Consider the function sequence $\overline{g}_i(\theta_{a(i)})$ given by Eq. (7-9) in the main text, we have $\overline{g}'_i(\theta_{a(i)}) = I_n - \left[\left(R + \nabla^2 L(\theta_{a*})\right)^{-1} R\right]^{i+1}$.*

*Proof.* According to Eq. (7-9) in the main text, we have

$$\overline{g}'_i(\theta_{a(i)}) = (R + \nabla^2 L(\theta_{a(i)}))^{-1} \times \left(\nabla^2 L(\theta_{a(i)}) + R\overline{g}'_{i-1}(\theta_{a(i)})\right) \\ - (R + \nabla^2 L(\theta_{a(i)}))^{-1} \times \left(\overline{g}^T_{i-1}(\theta_{a(i)}) \otimes I_n\right) \nabla^3 L(\theta_{a(i)}), \tag{26}$$

$$\overline{g}'_0(\theta_{a(i)}) = (R + \nabla^2 L(\theta_{a(i)}))^{-1}\nabla^2 L(\theta_{a(i)}) - (R + \nabla^2 L(\theta_{a(i)}))^{-1} \\ \times \left(\overline{g}^T_0(\theta_{a(i)}) \otimes I_n\right) \nabla^3 L(\theta_{a(i)}). \tag{27}$$

where $\otimes$ denotes the Kronecker product, $\nabla^3 L(\theta_{a(i)}) = \frac{d(\text{vec}(\nabla^2 L(\theta_{a(i)})))}{d\theta^T_{a(i)}}$ and $\nabla^2 L(\theta_{a(i)})$ is vectorized as $\text{vec}(\nabla^2 L(\theta_{a(i)}))$. Eq. (27) implies that every $\overline{g}'_0(\theta_{a(i)})$ contains $\nabla L(\theta_{a(i)})$ as a factor. Accordingly, due to $\nabla L(\theta_{a*}) = 0$, we have

$$\overline{g}_k(\theta_{a*}) = 0, \quad k = 0, 1, \cdots, N, \tag{28}$$

which, together with (26-27), yields

$$\bar{g}'_i(\theta_{a*}) = (R + \nabla^2 L(\theta_{a*}))^{-1}(\nabla^2 L(\theta_{a*}) + R\bar{g}'_{i-1}(\theta_{a*})), \tag{29}$$

$$\bar{g}'_0(\theta_{a*}) = (R + \nabla^2 L(\theta_{a*}))^{-1}\nabla^2 L(\theta_{a*}). \tag{30}$$

Based on Eq. (30), it is direct that

$$\bar{g}'_0(\theta_{a*}) = (R + \nabla^2 L(\theta_{a*}))^{-1}\nabla^2 L(\theta_{a*}) = I_n - (R + \nabla^2 L(\theta_{a*}))^{-1}R. \tag{31}$$

Then, assume for all $k \geqslant i+1$, there always holds

$$\bar{g}'_k(\theta_{a*}) = I_n - [(R + \nabla^2 L(\theta_{a*}))^{-1}R]^{k+1}. \tag{32}$$

Finally, we will show that in the case of $k = i$, (32) also holds.

$$\begin{aligned}
\bar{g}'_i(\theta_{a*}) &= (R + \nabla^2 L(\theta_{a*}))^{-1}\left(\nabla^2 L(\theta_{a*}) + R\bar{g}'_{i-1}(\theta_{a*})\right) \\
&= (R + \nabla^2 L(\theta_{a*}))^{-1}\left[\nabla^2 L(\theta_{a*}) + R\left(I_n - \left((R + \nabla^2 L(\theta_{a*}))^{-1}R\right)^i\right)\right] \quad (33) \\
&= I_n - \left[(R + \nabla^2 L(\theta_{a*}))^{-1}R\right]^{i+1}.
\end{aligned}$$

**Theorem 1.** *The weights of the actor network $\theta_a$ of OCPDQN can converge to the optimal weight $\theta_{a*}$.*

*Proof.* Based on Eq. (15) in the main text and Lemma 1, the following relationship can be obtained

$$\begin{aligned}
\theta_{a(i+1)} - \theta_{a*} &= \theta_{a(i)} - \theta_{a*} - \bar{g}_i(\theta_{a(i)}) \\
&= \theta_{a(i)} - \theta_{a*} - [\bar{g}_i(\theta_{a*}) + \bar{g}'_i(\theta_{a*})(\theta_{a(i)} - \theta_{a*}) + o(|\theta_{a(i)} - \theta_{a*}|)] \\
&\approx \theta_{a(i)} - \theta_{a*} - [\bar{g}'_i(\theta_{a*})(\theta_{a(i)} - \theta_{a*})] \\
&= (I - \bar{g}'_i(\theta_{a*}))(\theta_{a(i)} - \theta_{a*}) \\
&= \left(R + \nabla^2 \mathcal{L}(\theta_{a*})^{-1}R\right)^{i+1}\left(\theta_{a(i)} - \theta_{a*}\right).
\end{aligned}$$

where $o(|\theta_{a,c(i)} - \theta_{a*}|)$ is the higher order approximation term.

Since $\theta_{a*}$ is the minimizer of the loss function, we have $\nabla^2 \mathcal{L}(\theta_{a*}) > 0$. Therefore, $\rho\left(R + \nabla^2 \mathcal{L}(\theta_{a*})^{-1}R\right) < 1$. When the number of iterations $i$ is sufficiently large, $\left(R + \nabla^2 \mathcal{L}(\theta_{a*})^{-1}R\right)^{i+1} \to 0$, and thus we have

$$\theta_{a(i+1)} - \theta_{a*} = \left(R + \nabla^2 \mathcal{L}(\theta_{a*})^{-1}R\right)^{i+1}\left(\theta_{a(i)} - \theta_{a*}\right) = 0. \tag{34}$$

Hence, it can be seen that the actor network of OCPDQN ultimately converges to the optimal solution at a super-linear rate.

## A.2 CONVERGENCE ANALYSIS OF GN-OCPDQN

The weights of the actor network $\theta_a$ of GN-OCPDQN can converge to the optimal weight $\theta_{a*}$ at a super-linear rate. Based on Eq. (10) and Eq. (15) in the main text, we can get the following formulas:

$$\begin{aligned}
\theta_{a(i+1)} - \theta_{a*} &= \theta_{a(i)} - \theta_{a*} - \hat{g}_i(\theta_{a(i)}) \\
&= \theta_{a(i)} - \theta_{a*} - [\hat{g}_i(\theta_{a*}) + \hat{g}'_i(\theta_{a*})(\theta_{a(i)} - \theta_{a*}) + o(|\theta_{a(i)} - \theta_{a*}|)] \\
&\approx \theta_{a(i)} - \theta_{a*} - [\bar{g}'_i(\theta_{a*})(\theta_{a(i)} - \theta_{a*})] \\
&= (I - \hat{g}'_i(\theta_{a*}))(\theta_{a(i)} - \theta_{a*}) \\
&= \left(R + \nabla^2 \mathcal{L}(\theta_{a*})^{-1}R\right)^{i+1}\left(\theta_{a(i)} - \theta_{a*}\right) \quad (35) \\
&= \left(R + (J_r(\theta_{a*})^\top J_r(\theta_{a*}))^{-1}R\right)^{i+1}\left(\theta_{a(i)} - \theta_{a*}\right). \quad (36)
\end{aligned}$$

For parameters $\theta_{a(i)}$, there holds

$$\theta_{a(i+1)} - \theta_{a*} \leqslant \left(R + \nabla^2 \mathcal{L}(\theta_{a*})^{-1}R\right)^{i+1}\left(\theta_{a(i)} - \theta_{a*}\right). \tag{37}$$

Similarly, we have $J_r(\theta_{a*})^\top J_r(\theta_{a*}) > 0$, so $\left\| \left(R + J_r(\theta_{a*})^\top J_r(\theta_{a*})\right)^{-1} R \right\| < 1$. When the number of iterations $i$ is sufficiently large, $\left(R + \left(J_r(\theta_{a*})^\top J_r(\theta_{a*})\right)^{-1} R\right)^{i+1} \to 0$, we have

$$\theta_{a(i+1)} - \theta_{a*} = \left(R + (J_r(\theta_{a*})^\top J_r(\theta_{a*}))^{-1} R\right)^{i+1} \times \left(\theta_{a(i)} - \theta_{a*}\right) = 0. \tag{38}$$

The proof is complete.

### A.3 THE USE OF LARGE LANGUAGE MODELS

All LLM-generated suggestions and revisions were meticulously corrected and approved by the authors. This crucial step ensured that the final text remained fully aligned with the intended academic content and that no factual inaccuracies or conceptual distortions were introduced.

