# OpenReview forum: "Deep Q-Network Based on a New Optimization Method with Applications to Path Planning"
_ICLR.cc/2026/Conference — ICLR 2026 Conference Withdrawn Submission_

### Official Review · Reviewer_WVTh · 2025-10-23

**Soundness:** 1
**Presentation:** 1
**Contribution:** 1
**Rating:** 0
**Confidence:** 4

**Summary:**

This paper proposes two DQN variants for path planning that replace standard first-order optimization with second-order, control-inspired updates: OCPDQN (using Hessian-based updates) and GN-OCPDQN (using a Gauss–Newton approximation for efficiency). The authors claim improved convergence speed and greater stability compared to vanilla DQN and PPO. However, all experiments are limited to a single, simple 2D grid navigation task, and the theoretical arguments rely on assumptions that do not hold for standard DQN training. No statistical rigor or meaningful baselines are presented, and key methodological and reporting details are missing.

**Strengths:**

- The idea of using second-order or control-inspired optimization updates within a DQN framework is conceptually interesting and, if properly evaluated, could have practical merit.
- The paper attempts to provide theoretical motivation and an open-source code link.

**Weaknesses:**

- Extremely limited experimental scope: Only one simple 2D environment is used. There is no evaluation on standard RL or planning benchmarks, or any “complex environments” as claimed.
- No statistical rigor: All results are shown for a single seed. No means, error bars, or statistical tests are reported.
- Weak baselines: Comparison only to vanilla DQN and PPO. Missing all modern DQN variants and planning baselines.
- Theory-practice disconnect: Convergence theory assumes conditions (stationary targets, convexity) that are not present in practical DQN training.
- Figure quality: Plots are not publication-quality, lacking proper axes, legends, and clarity.
- robustness/generalization: No experiments with noise, randomness, or on procedurally generated layouts.
- Ablations and analysis: No ablations or sensitivity analysis for key design choices.

**Questions:**

1. What exact assumptions are required for your convergence proofs to apply? How do these relate to the real DQN training loop with bootstrapped targets?
2. Why do you only evaluate on a single, handcrafted 2D map? Why not use standard RL or planning benchmarks to validate claims of broad applicability?
3. Why are strong DQN variants and classical planning baselines not included in the experiments?
4. How do the methods perform on random or procedurally generated maps, or in the presence of noise/stochasticity?

---

### Official Review · Reviewer_Ttrv · 2025-10-30

**Soundness:** 1
**Presentation:** 2
**Contribution:** 3
**Rating:** 2
**Confidence:** 3

**Summary:**

This paper introduces two optimization frameworks for Deep Q-Networks (DQN), namely OCPDQN and GN-OCPDQN, by formulating the training process as an optimal control problem. In OCPDQN, the authors use a novel recursive update rule based on second-order information (the Hessian matrix), while GN-OCPDQN applies a Gauss-Newton approximation to reduce computation cost. The goal is to accelerate convergence compared to standard gradient descent approaches used in deep reinforcement learning. The methods are evaluated on a custom 2D path planning task, demonstrating a substantial reduction in training episodes required to reach near-optimal performance when compared to vanilla DQN and PPO.

**Strengths:**

The paper presents an interesting approach by connecting DQN optimization to optimal control theory. Preliminary results suggest faster convergence of both the presented algorithms with respect to standard DQN and PPO, though on a very limited benchmark. The use of a Gauss-Newton approximation in GN-OCPDQN allows limiting the computational cost associated with the computation of the Hessian matrix. The authors provide anonymized code, aiding reproducibility. The overall structure and argument flow are clear and logical, moving systematically from motivation to algorithms to experiments.

**Weaknesses:**

## Theoretical proofs
The convergence proofs in Sections 4.1.3, 4.2.3, and in the Appendix are unclear, and seem to contain errors.
* Equation (10) performs Taylor expansion:
    $$g_i(\theta_a(i)) \approx g_i(\theta_{a*}) + g_i'(\theta_{a*})(\theta_a(i) - \theta_{a*}) + o(|\theta_a(i) - \theta_{a*}|)$$
    and then assumes $g_i(\theta_{a*}) = 0$ without proof. However, from Equation (8):
    $$g_i(\theta_{a*}) = [R + \nabla^2 L(\theta_{a*})]^{-1}[0 + Rg_{i-1}(\theta_{a*})] = [R + \nabla^2 L(\theta_{a*})]^{-1}Rg_{i-1}(\theta_{a*})$$
    This is not necessarily zero and depends recursively on $g_{i-1}(\theta_{a*})$. The proof in Appendix A.1 (Lemma 1, Eq. 28) commits circular reasoning by confusing $\bar{g_0}'(\theta_{a*})$ (derivative) with $\bar{g_0}(\theta_{a*})$ (function value). Without establishing $g_i(\theta_{a*}) = 0$ for all $i \geq 0$, the convergence bound in Equation (11) is meaningless, and the claimed super-linear convergence rate is unsubstantiated.
* The paper contains notation inconsistencies: Section 4.1.1 uses $g_i(\theta_a(i))$, Appendix A.1 suddenly uses $\bar{g_i}(\theta_a(i))$ and $\hat{g_i}(\theta_a(i))$, without explaining the differences between the three.
* Multiple formulas misplace inverses (e.g., $(R+\nabla^2\mathcal{L}(\theta_{a*}))^{-1}R$).

Additionally, the transition function defined in Section 3.2.1 $\mathcal{F}(s_t, c_t, a_t)$ contains $c_t$, which is never introduced.

## Experiments
* Experiments are restricted to a single 2D grid-world environment (100 × 100 meter grid) with a single obstacle configuration. This is insufficient to support the claims of "higher robustness" and drastic reduction of computation time "particularly in high-dimensional settings". The authors should perform ablations with increasing environment size, different obstacle configurations, and .
* All plots lack error bars or confidence intervals, suggesting just a single run per configuration was performed. The authors should clarify this aspect. In order to support the claims, it is necessary to perform experiments using multiple random seeds and to perform variance analysis on the learning curves.
* The claimed "$47\%$ faster convergence" is misleading, as it refers only to episode count, not wall-clock time. The authors' own Figure 4 shows GN-OCPDQN has an average step time of $0.35s$ versus $0.02s$ for standard DQN. Without total training time comparisons, the practical benefit is unclear. Also, no step time values are reported for OCPDQN.
* The authors just compare against PPO and standard DQN. They should consider comparing at least against [1].

\noindent In conclusion, the paper's key theoretical and empirical claims are not convincingly supported due to apparent errors in the proofs, insufficient experiments, and lack of clarity in both notation and methodology. Without substantial revisions addressing these issues, the current submission does not meet the standard for acceptance at ICLR.

## References
[1] Beltiukov, R. (2020). Optimizing Q-Learning with K-FAC Algorithm. In: van der Aalst, W., et al. Analysis of Images, Social Networks and Texts. AIST 2019. Communications in Computer and Information Science, vol 1086. Springer, Cham. https://doi.org/10.1007/978-3-030-39575-9\_1

**Questions:**

* Can you clarify all notational inconsistencies in your derivations and the reason why $g_i(\theta_{a*}) = 0$?
* Did you perform experiments using multiple random seeds? Would you be able to add error bars in your plots?
* How do step times and computational cost scale for your algorithms in higher-dimensional settings (bigger environments, but also bigger networks) as claimed?
* Can you clarify the role of $c_t$ in the transition function and ensure all variables are precisely and consistently defined?
* How does your proposed methods relate to natural gradient or K-FAC approaches, which also precondition gradients using curvature information?

---

### Official Review · Reviewer_gyPF · 2025-10-31

**Soundness:** 2
**Presentation:** 2
**Contribution:** 1
**Rating:** 0
**Confidence:** 4

**Summary:**

This paper proposes two frameworks, OCPDQN and GN-OCPDQN, which integrate optimal control principles and Gauss–Newton optimization into the Deep Q-Network (DQN) framework for path planning tasks. The authors claim that their methods achieve super-linear convergence rates and faster training compared to standard DQN and PPO baselines. While the paper is clearly structured and mathematically detailed, the core methodology--using second-order optimization for DQN--is not novel. Furthermore, the experimental validation is limited to a simple 2D grid environment, and the claimed contributions overlap significantly with long-established research in reinforcement learning and optimal control theory.

**Strengths:**

The paper presents a clear and well-organized structure, with detailed algorithmic formulations and pseudo-code for both OCPDQN and GN-OCPDQN. The inclusion of convergence proofs adds formal rigor, and the experiments, though limited, are well visualized and include multiple performance metrics such as reward progression, path length, and step time. The overall writing is coherent and readable, and the results show measurable improvements in convergence speed.

**Weaknesses:**

The main weakness lies in the lack of novelty. The proposed methods essentially repackage existing second-order and natural-gradient optimization ideas within a DQN framework, a line of research extensively explored in prior works such as Natural Policy Gradient (Kakade, 2001), Hessian-free Optimization (Martens, 2010), and TRPO (Schulman, 2015). The theoretical connection between OCP-based formulations and reinforcement learning remains superficial and does not extend beyond restating known second-order update rules.
Experimentally, the evaluation is performed in a simple static 2D grid world, far from the complexity typically required for ICLR-level contributions. The paper also fails to compare against stronger and more recent baselines such as SAC, TD3, or A3C, limiting the generality of its conclusions.

**Questions:**

-

**Details Of Ethics Concerns:**

A major ethical concern involves citation and attribution practices. The paper consistently fails to cite the original sources of foundational algorithms such as A* (Hart et al., 1968), RRT (LaValle, 1998), Genetic Algorithms (Holland, 1975), and Particle Swarm Optimization (Kennedy & Eberhart, 1995). Instead, it repeatedly cites recent derivative or survey papers--mostly from 2022–2025 and predominantly authored by researchers from a particular country--as substitutes for the original works. This pattern suggests deliberate replacement rather than oversight and constitutes a citation misconduct or derivative citation violation.

In addition, the paper misrepresents long-established reinforcement learning optimization methods, e.g., Gauss–Newton, Natural Gradient, Hessian-free approaches, as novel contributions without referencing seminal works by Kakade, Martens, or Schulman. This leads to a misrepresentation of prior art and an inflation of claimed originality.

Taken together, these issues raise serious concerns regarding the paper’s scholarly integrity and compliance with standard ethical citation practices expected at ICLR. A full citation audit and major revision are required before any further consideration.

---

### Official Review · Reviewer_dDwC · 2025-11-04

**Soundness:** 3
**Presentation:** 3
**Contribution:** 3
**Rating:** 6
**Confidence:** 4

**Summary:**

This paper addresses traditional Deep Q-Network (DQN) limitations in path planning (slow convergence, local optima susceptibility) by proposing two frameworks: OCPDQN (fusing DQN with Optimal Control Problem, OCP) and GN-OCPDQN (adding Gauss-Newton approximation to avoid Hessian computation). Both achieve super-linear convergence, with OCPDQN converging 47%/40% faster than DQN/PPO and GN-OCPDQN 41%/33% faster in 2D static path planning tests. The key contributions are:
1. Propose two novel frameworks (OCPDQN, GN-OCPDQN) tailored for path planning, addressing DQN’s optimization bottlenecks.
2. Integrate OCP into DQN training to significantly reduce iteration counts, and prove both frameworks’ super-linear convergence.
3. Introduce Gauss-Newton approximation in GN-OCPDQN to eliminate Hessian computation, improving scalability for high-dimensional networks.

**Strengths:**

Originality is strong. The paper targets DQN’s core flaw (gradient descent-induced slow convergence) by innovatively fusing OCP—a super-linearly convergent optimization method—into DQN parameter updates, a design rarely seen in path-planning DQN variants. It further addresses OCP’s Hessian computation overhead with GN-OCPDQN’s Gauss-Newton approximation, making the framework adaptable to high-dimensional networks and distinguishing it from naive OCP-DQN combinations.

Quality is good. The work is theoretically rigorous, with detailed Appendix proofs for both frameworks’ super-linear convergence, leveraging Lemmas to validate optimal weight convergence. Experimentally, it uses specific hardware (Intel Xeon Gold 6348, NVIDIA A800) and clear hyperparameters (ε decay, discount factor 0.99), with quantitative speedups and qualitative path comparisons (Fig. 3) supported by representative baselines (DQN for value-based, PPO for policy gradient methods).

Clarity is good. The paper follows a logical “challenge→preliminaries→method→experiments” structure, with pseudocodes (Alg. 1/2) detailing training loops and Figures 2–4 intuitively presenting convergence, path quality, and step time. Mathematical notation (e.g., θₐ for actor parameters) is consistent, and key terms (OCP rules, GN approximation) are explained on first mention to avoid ambiguity.

Significance is high. It solves a practical path-planning pain point: DRL’s slow convergence limits real-world deployment in robotics/autonomous driving. The frameworks balance speed (OCP) and efficiency (GN), and their optimization logic is transferable to other DRL tasks, offering both research (optimization innovation) and engineering (deployment feasibility) value.

**Weaknesses:**

1. Experimental environment is overly simplistic (only static 2D 100×100m map, no dynamic obstacles/3D scenarios).

Suggestion: Add dynamic obstacle tests (e.g., moving obstacles) or 3D tasks to verify robustness.

2. Baselines are limited (only vanilla DQN and PPO; missing Dueling DQN/Double DQN).

Suggestion: Include state-of-the-art DQN variants to highlight advantages over existing optimizations.

3. No analysis of control matrix R’s impact (R affects OCP convergence but lacks sensitivity tests).

Suggestion: Test R values (0.1, 1, 10) and report optimal selection criteria.

4. Real-time performance only measures step time, not end-to-end inference latency.

Suggestion: Test end-to-end latency (input state→action output) for deployment relevance.

**Questions:**

1. What criteria guided control matrix R selection in OCPDQN? How do you balance R’s impact on convergence speed and stability?

2. In GN-OCPDQN, how is the residual Jacobian computed during training? Does this add significant overhead vs. OCPDQN’s Hessian calculation?

3. Have you tested the frameworks in dynamic obstacle environments? If not, what modifications would be needed?

4. Why only compare with vanilla DQN and PPO? How would your frameworks perform against improved DQNs (e.g., Dueling DQN)?

---

### Note · Authors · 2025-11-14

I have read and agree with the venue's withdrawal policy on behalf of myself and my co-authors.